# Identification of ERAD-dependent degrons for the endoplasmic reticulum lumen

**Rachel Sharninghausen[1†], Jiwon Hwang[1†], Devon D Dennison[2], Ryan D Baldridge[1,2]***

[1]Department of Biological Chemistry, University of Michigan Medical School, Ann Arbor, United States; [2]Cellular and Molecular Biology Program, University of Michigan Medical School, Ann Arbor, United States

## eLife Assessment

This **important** study identifies a short amino acid sequence that, when fused in multimeric form to the amino termini of luminal ER proteins, initiates proteasomal degradation via the Hrd1 ER quality control ubiquitin ligase complex. The authors provide **solid** evidence that this sequence functions as a "degron" for ER proteins. Future work is required to obtain a more detailed view of the properties of this degron, the mechanisms underlying its recognition by ER-resident and cytoplasmic factors, and the in vivo relevance of the findings.

**\*For correspondence:**
ryanbald@umich.edu

[†]These authors contributed equally to this work

## Abstract

Degrons are minimal protein features that are sufficient to target proteins for degradation. In most cases, degrons allow recognition by components of the cytosolic ubiquitin proteasome system. Currently, all of the identified degrons only function within the cytosol. Using *Saccharomyces cerevisiae*, we identified the first short linear sequences that function as degrons from the endoplasmic reticulum (ER) lumen. We show that when these degrons are transferred to proteins, they facilitate proteasomal degradation through the endoplasmic reticulum associated degradation (ERAD) system. These degrons enable degradation of both luminal and integral membrane ER proteins, expanding the types of proteins that can be targeted for degradation in budding yeast and mammalian tissue culture. This discovery provides a framework to target proteins for degradation from the previously unreachable ER lumen and builds toward therapeutic approaches that exploit the highly conserved ERAD system.

## Introduction

Protein degradation plays an essential role in regulating diverse cellular processes including cellular signaling, metabolic adaptation, and cell cycle regulation. The ubiquitin proteasome system is the primary cellular degradation route, accounting for over 80% of protein degradation (***Collins and Goldberg, 2017***). Ubiquitination requires the concerted action of the ubiquitination cascade comprising E1 ubiquitin activating enzymes, E2 ubiquitin conjugating enzymes, and E3 ubiquitin ligases. For ubiquitin, in *Saccharomyces cerevisiae* there is 1 E1, 11 E2s, and >60 E3s whereas mammals have 2 E1s, ~40 E2s, and >600 E3s (***Clague et al., 2015***). The specificity of the ubiquitination process is primarily driven by the E3 ubiquitin ligases and a major challenge in ubiquitin biology is the identification of the sequences that target proteins for degradation (called 'degrons').

Degrons are usually short linear motifs and, by definition, are sufficient to confer degradation when transferred to otherwise stable proteins. Degrons can be acquired, or inherent. Acquired degrons are

generally post-translational modifications that can be based on proteolytic cleavage, phosphorylation, or acetylation. Inherent degrons are features of the primary polypeptide sequence formed by linear or conformational epitopes. Inherent degrons can be shielded when proteins are appropriately folded or incorporated into larger protein complexes. The first degrons to be discovered were at the amino-terminus of proteins (*Bachmair et al., 1986*) and these N-degrons were eventually summarized as the 'N-end rule' (*Varshavsky, 2011*), and later the 'N-degron pathways' (*Varshavsky, 2019*). Recent systems-level analyses have expanded the availability of known degrons broadly (*Geffen et al., 2016*; *Mashahreh et al., 2022*), with a series of 'C-end rules' (*Koren et al., 2018*; *Lin et al., 2018*), and additional variations of the N-degron pathways (reviewed in *Timms and Koren, 2020*). Even with a wide range of physiological roles for protein degradation, degrons are still unidentified for most E3 ubiquitin ligases. All known degrons target cytosolic proteins for ubiquitination and degradation by the proteasome and the identification of degrons for a few key ubiquitin ligases has enabled exploitation of the proteasome to facilitate targeted protein degradation of 'undruggable' cytosolic proteins (*Samarasinghe and Crews, 2021*). However, many proteins that originate from the lumen of membrane-encapsulated organelles are also degraded using the proteasome and degrons for these organelles are mysterious (*Christianson and Carvalho, 2022*). Therefore, targeting proteins for degradation from within organelles requires a detailed understanding of local organellar protein quality control systems.

The endoplasmic reticulum (ER) represents the organelle with the largest flux of proteins, with over 40% of proteins translocated into the ER before trafficking to other organelles or secretion from the cell. Both soluble luminal proteins and integral membrane proteins are folded in the ER and undergo quality control before being released into the secretory pathway. At the ER, the primary protein quality control pathways are, collectively, referred to as endoplasmic reticulum associated degradation (ERAD) and these systems are highly conserved among all eukaryotes (*Huyer et al., 2004*; *Vashist and Ng, 2004*; *Carvalho et al., 2006*; *Denic et al., 2006*; *Foresti et al., 2014*; *Khmelinskii et al., 2014*). One such system, the Hrd1-centric ERAD complex, recognizes proteins not passing quality control and retrotranslocates them from the ER lumen to the cytosol for ubiquitin-mediated proteasomal degradation. In *S. cerevisiae*, the Hrd1 complex comprises five proteins: Hrd1, Hrd3, Usa1, Der1, and Yos9. In the cytosol, this pathway requires a highly conserved AAA-ATPase (Cdc48), its cofactors (Ufd1 and Npl4), and the ubiquitination proteasome system to degrade ERAD substrates (*Bays et al., 2001*; *Ye et al., 2001*; *Jarosch et al., 2002*; *Rabinovich et al., 2002*). Soluble, luminal ERAD substrates are retrotranslocated by hetero-oligomers of Hrd1/Der1 (*Mehnert et al., 2014*; *Wu et al., 2020*; *Pisa and Rapoport, 2022*), or in some cases, homo-oligomers of Hrd1 (*Carvalho et al., 2010*; *Baldridge and Rapoport, 2016*; *Schoebel et al., 2017*). Hrd1 is sufficient for the basic functions of ERAD but without the other complex components, loses the specificity that normally defines the system (*Denic et al., 2006*). This system has broad specificity and seems to be able to distinguish folded and unfolded proteins (*Stein et al., 2014*). Despite nearly three decades of study, degrons (neither sequences nor features) that allow degradation through the Hrd1-centric ERAD pathway remain a complete mystery (*Needham et al., 2019*).

Here, we have identified the first short luminal degrons of the Hrd1-centric ERAD pathway. We demonstrate that a degron can be functionalized to drive protein degradation from the ER lumen, a previously inaccessible cellular location. We show that this technology can drive degradation of both soluble, luminal ER proteins and integral membrane ER proteins. This degron works in both budding yeast and mammalian tissue culture. This work provides an exciting and simple method of targeting proteins for degradation from within the ER by exploiting the highly conserved ERAD system.

## Results
### Identification of ER-localized degrons

The ER is the primary location for protein quality control within the secretory pathway, but how ER protein quality control systems distinguish folded from unfolded proteins is unclear. We wanted to understand the degrons recognized within the ER lumen and started by designing an ER-targeted reporter of protein stability. We targeted a tandem fluorescent protein timer (tFT) to the ER to function as a reporter of protein stability (ER-tFT, *Figure 1A* and *Figure 1—figure supplement 1A*). tFTs contain a fast-maturing fluorescent protein (here, superfastGFP; *Fisher and DeLisa, 2008*) and a

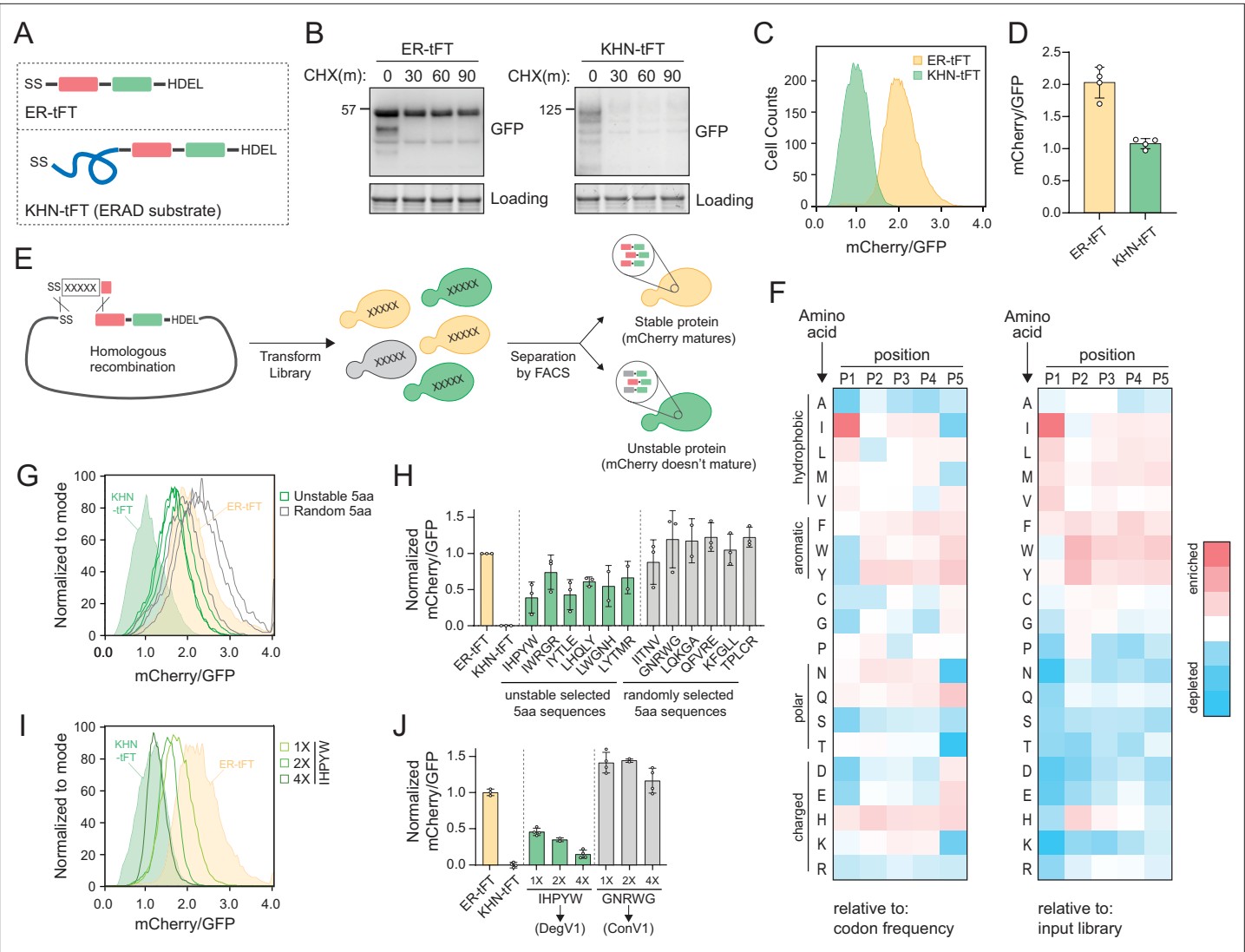

**Figure 1.** Identification of endoplasmic reticulum (ER)-localized degrons. (**A**) Schematic depicting the ER-tandem fluorescent protein timer (tFT) and KHN-tFT constructs, which contain an ER-targeting signal sequence (SS), mCherry (red), superfastGFP (green), and the HDEL ER retention sequence. KHN-tFT functions as a quickly degraded ER associated degradation (ERAD) substrate (a positive control for degradation). (**B**) Wild-type yeast expressing the constructs described in (**A**) were treated with cycloheximide (CHX) for 0, 30, 60, or 90 min, harvested, and protein levels were assessed by immunoblotting using anti-GFP antibodies. Total protein was visualized in gel using stain-free technology (Loading). (**C**) Flow cytometry of yeast expressing the constructs in (**A**) treated with CHX for 2 hr. The mCherry/GFP fluorescence intensity ratio of each cell was calculated and plotted. (**D**) Quantification of the mean mCherry/GFP ratio of four biological replicates as in (**C**). (**E**) Overview of the pentapeptide library generation and isolation of unstable variants using fluorescence-activated cell sorting (FACS). A DNA fragment containing the pentapeptide-ER-tFT library was electroporated with linearized ER-tFT plasmid. The resulting yeast library contains a mixture of variants that are separated by FACS, with less stable variants having decreased mCherry/GFP fluorescence intensity compared to stable variants. (**F**) Heatmap of amino acid enrichments at each position within the unstable pentapeptide library. Values are displayed relative to either codon usage (left) or relative to the input library (right). (**G**) As in (**C**) with strains expressing either ER-tFT (yellow fill), KHN-tFT (green fill), individual FACS isolates from the 'unstable' pentapeptide sequences (green lines), or randomly selected pentapeptide-ER-tFT sequences from the input library (gray lines). (**H**) Quantification of at least two biological replicates conducted as in (**G**). The unstable groups (green) and random groups (gray) were significantly different from each other using a one-way ANOVA and Tukey's multiple comparisons tests. (**I**) As in (**C**) with strains expressing either ER-tFT (yellow fill), KHN-tFT (green fill), a single IHPYW (1X), 2X repeat of IHPYW (2X), or 4X repeat of IHPYW at the N-terminus of ER-tFT. (**J**) Quantification of two to three biological replicates of (**I**).

The online version of this article includes the following source data and figure supplement(s) for figure 1:

**Source data 1.** Uncropped and labeled gels for *Figure 1*.

**Source data 2.** Raw unedited gels for *Figure 1*.

**Source data 3.** Numerical source data for plots displayed in *Figure 1*.

**Figure supplement 1.** Optimization of the experimental design.

slower-maturing fluorescent protein (here, mCherry; *Shaner et al., 2004*) and have been used effectively to identify N-terminal degrons that function in the cytosol and nucleus (*Khmelinskii et al., 2012*; *Kats et al., 2018*). By measuring the ratio of mCherry to GFP fluorescence, a protein's stability can be assessed; the lower the ratio, the more unstable the protein. To test whether the ER-tFT could successfully distinguish stable from unstable proteins, we compared the ER-tFT to a well-characterized, unstable, luminal ERAD substrate, KHN, tagged with the tFT (KHN-tFT, *Figure 1A*). Using a cycloheximide chase followed by immunoblotting, we found the ER-tFT was quite stable, while the KHN-tFT was degraded with a half-life of less than 30 min (*Figure 1B*), consistent with previous reports (*Vashist et al., 2001*). Using flow cytometry, the two proteins were also distinguishable following cycloheximide treatment (*Figure 1C and D* and *Figure 1—figure supplement 1B*). After establishing the tFT reporter could distinguish protein stability within the ER, we turned our attention to identifying luminal degrons.

To identify degrons that function within the ER lumen, we generated a library of short, linear peptide sequences embedded into the ER-tFT. Using PCR with degenerate primers, we generated an unbiased pentapeptide library encoded in a DNA fragment to use for homologous recombination in cells (*Figure 1E*, *Figure 1—figure supplement 1C and D*). The theoretical amino acid diversity of a pentapeptide library is 3.2 million ($20^5$). We transformed the library into wild-type yeast and obtained 1.2 million transformants. Using fluorescence-activated cell sorting we separated cells expressing the pentapeptide-ER-tFT by their mCherry/GFP ratio (*Figure 1—figure supplement 1E and F*). We collected an 'unstable' bin (exhibiting a low mCherry/GFP ratio), which encompassed 4% of the sorted cells, for sequencing (*Figure 1—figure supplement 1F*).

The sequencing results of the unstable sorted bin illuminated an enrichment of pentapeptides beginning with isoleucine and leucine (*Source data 1*). Specifically, isoleucine at the first position was present in 17.7% of unstable pentapeptides while leucine was present in 14.8%. Based on codon usage in a random sampling, isoleucine was predicted to appear 4.7% of the time (3/64 codons) giving a 3.8-fold enrichment in our dataset. Leucine was predicted to appear at a specific position 9.4% of the time (6/64 codons) giving a 1.6-fold enrichment in our dataset (*Figure 1F*). When compared to the input library abundance, enrichment corresponds to 2.7- and 1.5-fold, respectively (*Figure 1F*). Rather than attempting to build a consensus sequence that would have to report on many steps in ER quality control, we selected pentapeptide sequences present in the unstable bins that broadly represented the enrichment trends we observed to individually clone and characterize (*Figure 1F* and *Source data 1*).

We compared the stability of six different pentapeptides with isoleucine or leucine at position one to ER-tFT alone, to KHN-tFT, and to a set of randomly selected pentapeptides. KHN-tFT was the least stable, followed by pentapeptides selected from the unstable bin and containing isoleucine or leucine at position one. Several of the randomly selected pentapeptides exhibited a slight reduction in stability, relative to ER-tFT alone, but pentapeptides selected from the unstable bin were consistently less stable (*Figure 1G and H*, and *Figure 1—figure supplement 1G*). Encouraged by our results, we sought to find a peptide sequence that would match KHN-tFT instability. We found that repeats of IHPYW, one of the most unstable sequences, dramatically decreased protein stability, with a 20 amino acid 4X(IHPYW) repeat successfully resembling the KHN control (*Figure 1I and J*). On the other hand, simply repeating a stable control sequence (GNRWG) did not destabilize the protein (control variant 1 [ConV1], *Figure 1J*). Based on these results, we concluded that the 4X(IHPYW) sequence, which we called DegV1 (Degron Variant 1), functions as an ER-localized degron.

## DegV1 is an ERAD-dependent degron degraded by the proteasome

We next tested whether DegV1 functioned as an ER luminal degron by transferring it from the original ER-tFT reporter (*Figure 1*) to a second protein known to fold in the ER lumen. We targeted the LaG16 anti-GFP nanobody (*Fridy et al., 2014*) to the ER using the mating factor alpha signal sequence and an ER retention signal (ER-NbGFP, *Figure 2—figure supplement 1A*). First, to ensure the signal sequence was removed and did not contribute to the DegV1 degron, we compared the molecular weight of a cytosolic NbGFP (without signal sequence), a cytosolic DegV1-NbGFP (without signal sequence), and the ER-DegV1-NbGFP (with signal sequence) (*Figure 2A*). Based on the size of the ER-DegV1-NbGFP compared to DegV1-NbGFP, we concluded that ~90% of the ER-targeted construct had the signal sequence removed and uncleaved signal sequence was unlikely to factor into

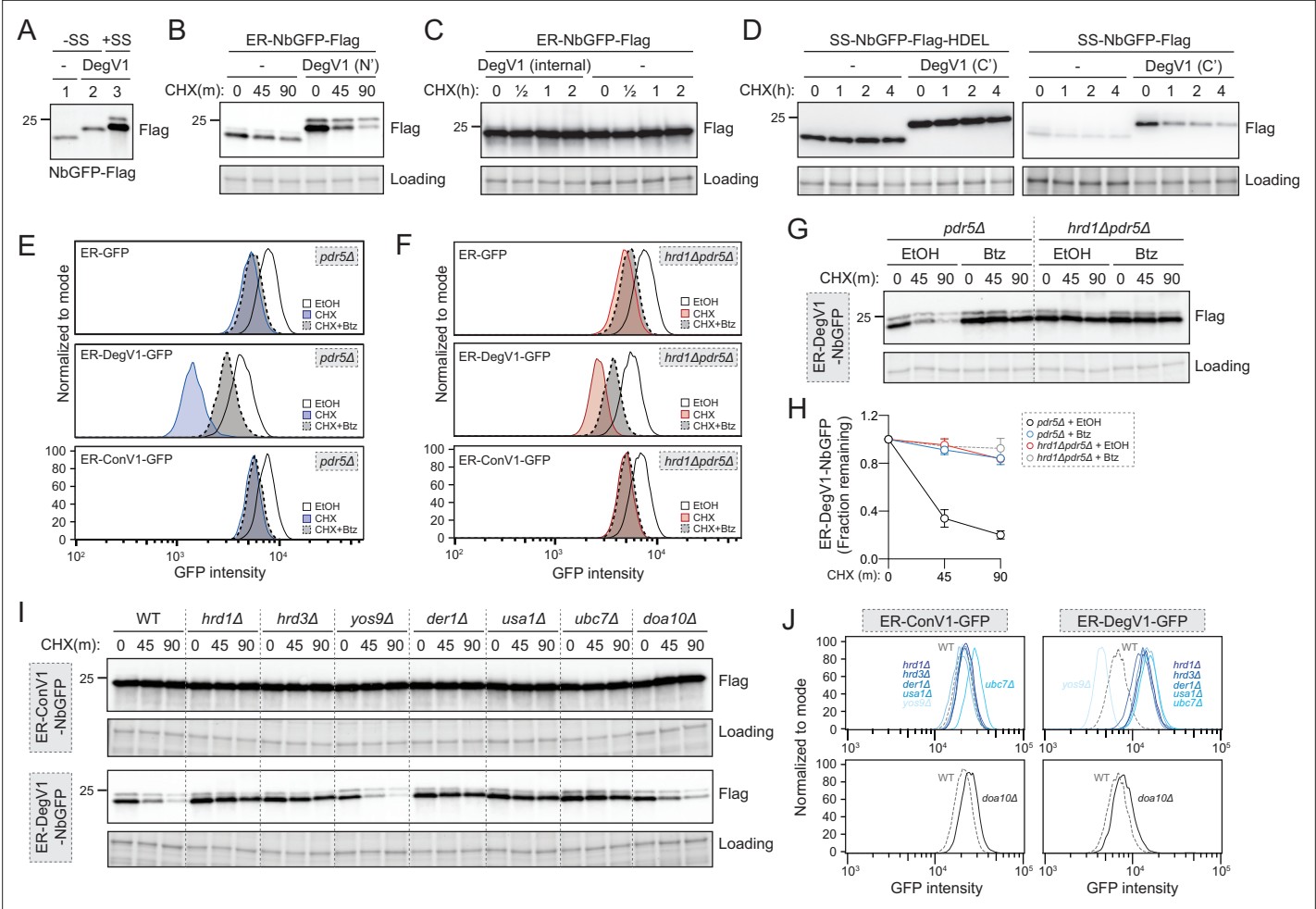

**Figure 2.** DegV1 is an endoplasmic reticulum associated degradation (ERAD)-dependent degron degraded by the proteasome. (**A**) The apparent molecular weight of anti-GFP nanobodies were assessed (NbGFP-Flag) either with or without signal sequence, in the presence or absence of DegV1, were assessed by SDS-PAGE electrophoresis followed by immunoblotting with anti-Flag antibody. This panel is representative of two biological replicates. (**B**) The degradation of ER-targeted anti-GFP nanobodies (ER-NbGFP-Flag) either with, or without, DegV1 were monitored following addition of cycloheximide (CHX), using SDS-PAGE and immunoblotting. Loading controls were visualized by stain-free technology. (**C**) The degradation of ER-NbGFP-Flag with DegV1 replacing the CDR3 region was analyzed as in (**B**). (**D**) The degradation of a nanobody with DegV1 located either directly preceding the C-terminal HDEL ER retention signal (left) or directly at the C-terminus of the nanobody (right) was analyzed as in (**B**). (**E**) Degradation of the ER-targeted proteins ER-GFP (top panel), ER-DegV1-GFP (middle panel), or ER-ConV1-GFP (bottom) were analyzed in a *pdr5Δ* strain by flow cytometry following either ethanol (EtOH) or CHX treatment for 2 hr. Where indicated, cells were pretreated with bortezomib (Btz) for 2 hr. (**F**) As in (**E**), but in a *hrd1Δpdr5Δ* strain. (**G**) Degradation of ER-DegV1-NbGFP was followed in *pdr5Δ or hrd1Δpdr5Δ* strain as in (**B**). Where indicated, cells were pretreated with Btz for 2 hr. (**H**) Quantification of (**G**) with error bars representing the standard deviation. (**I**) Degradation of ER-ConV1-NbGFP (top panel) or ER-DegV1-NbGFP (bottom panel) was followed in of ERAD component deletion strains as in (**B**). This panel is representative of two independent biological replicates. (**J**) The degradation of ER-ConV-GFP (left), or ER-DegV-GFP (right), were analyzed in the indicated ERAD component deletion strains by flow cytometry following cycloheximide treatment for 2 hr. All panels in this figure are representative of at least three independent biological replicates, unless otherwise indicated.

The online version of this article includes the following source data and figure supplement(s) for figure 2:

**Source data 1.** Uncropped and labeled gels for *Figure 2*.

**Source data 2.** Raw unedited gels for *Figure 2*.

**Source data 3.** Numerical source data for plots displayed in *Figure 2*.

**Figure supplement 1.** Construct layout and degradation experiments.

**Figure supplement 1—source data 1.** Uncropped and labeled gels for *Figure 2—figure supplement 1*.

**Figure supplement 1—source data 2.** Raw unedited gels for *Figure 2—figure supplement 1*.

**Figure supplement 2.** DegV1 is degraded in the cytoplasm.

*Figure 2 continued on next page*

*Figure 2 continued*

**Figure supplement 2—source data 1.** Uncropped and labeled gels for *Figure 2—figure supplement 2*.

**Figure supplement 2—source data 2.** Raw unedited gels for *Figure 2—figure supplement 2*.

**Figure supplement 2—source data 3.** Numerical source data for plots displayed in *Figure 2—figure supplement 2*.

the function of DegV1 (*Figure 2A*). Using a cycloheximide chase, the ER-NbGFP protein alone was quite stable, but embedding DegV1 after the N-terminal signal sequence destabilized ER-NbGFP with a half-life of approximately 30 min (*Figure 2B*). This result validated DegV1 as the first, relatively short, degron sequence that can be used for targeting ER luminal substrates for degradation. Next, we tested if DegV1 could also function as an internal or C-terminal degron. To test whether the DegV1 sequence worked within internal loops, we used the same nanobody scaffold but replaced the complementary determining region 3 with DegV1 and found that the presence of an internal DegV1 made no difference in protein stability when compared to the nanobody alone (*Figure 2C* and *Figure 2—figure supplement 1A*). Testing DegV1 at the C-terminus of the NbGFP was complicated by the requirement for an ER retention signal. We integrated the DegV1 sequence immediately before the HDEL retention signal, and, again, this resulted in no change in the stability of the NbGFP target protein (*Figure 2D* and *Figure 2—figure supplement 1A*). The ER retention signal is an important component of this construct's innate stability, because loss of the HDEL signal, with or without the DegV1 sequence at the extreme C-terminus, results in degradation in the vacuole or secretion from the cell (*Figure 2D*). Together, these results suggest that the DegV1 degron was capable of acting as an ER luminal degron, but only when positioned at the N-terminus of ER-localized proteins.

DegV1 was able to actively target proteins for degradation from the ER lumen. The majority of protein degradation occurs at the proteasome, so we next tested whether DegV1-tagged proteins were degraded by the proteasome. We confirmed ER-GFP was ER-localized (*Figure 2—figure supplement 1B*) and examined the stability of the fluorescent ER-localized construct GFP alone, or with DegV1 or ConV1 in cells treated for 2 hr with the proteasomal inhibitor bortezomib and/or cycloheximide. The stability of ER-GFP alone was similar either in the presence (dashed outline) or absence (solid outline) of bortezomib (panel 1, *Figure 2E*). Appending DegV1 immediately after the signal sequence resulted in degradation of ER-GFP (solid blue line), which was inhibited by adding bortezomib (dashed line, panel 2, *Figure 2E*). This indicated DegV1 targets the luminal ER-GFP for proteasomal degradation. As expected, appending a control sequence (ConV1) of the same length was similarly stable to ER-GFP alone and stability was not affected by bortezomib (solid line versus dashed line, panel 3, *Figure 2E*, see also *Figure 2—figure supplement 1C*). Therefore, DegV1-targeted luminal ER substrates were degraded by the proteasome.

Proteasomal degradation of luminal ER proteins is mediated by the Hrd1-ERAD system (*Bordallo et al., 1998*). Consequently, we suspected that Hrd1-centered ERAD mediates DegV1-targeted proteasomal degradation. Again, we tested the stability of ER-GFP in cells lacking the central component to the Hrd1-ERAD system, the ubiquitin ligase Hrd1. The steady-state levels of ER-GFP alone or with ConV1 were similar in the absence of Hrd1 and remained unaffected by proteasome inhibition (panels 1 and 3, *Figure 2F*). In contrast, DegV1-containing ER-GFP was more stable in an *hrd1Δ* strain, and bortezomib resulted in little further stabilization of DegV1-containing GFP (panel 2, *Figure 2F*, see also *Figure 2—figure supplement 1D*), possibly from a small fraction being incompletely translocated into the ER. In the absence of Hrd1, we observed some ER leakage of ER-DegV1-GFP to the vacuole and appearance of a degradation resistant GFP fragment that resembled the known luminal ERAD substrate, CPY*-GFP (indicated by * in *Figure 2—figure supplement 1E and F*). Altogether, these results are consistent with a role for Hrd1 in the degradation of ER-DegV1-GFP.

To further test the role of Hrd1 in DegV1-targeted degradation of luminal ER substrates, we tested additional soluble ER proteins (*Figure 2G* and *Figure 2—figure supplement 1G*) that were, otherwise, relatively stable in the ER lumen (*Figure 2—figure supplement 1G*). As expected, when DegV1 was appended to the ER-NbGFP, we found the protein was unstable, with a half-life of approximately 30 min in a cycloheximide chase (*Figure 2G and H*, and *Figure 2—figure supplement 1G*). Next, we tested whether other known components of the Hrd1 ERAD complex were required for DegV1 degradation. We found that, with the notable exception of Yos9, the previously identified components of the ERAD-L complex (Hrd1, Hrd3, Der1, and Usa1) were required for DegV1 degradation (*Figure 2I*

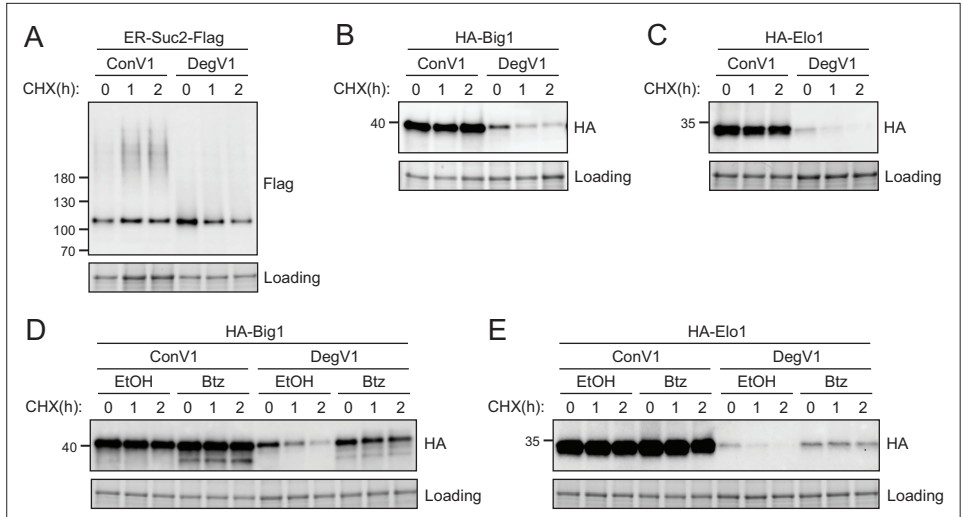

**Figure 3.** DegV1 targets endogenous endoplasmic reticulum (ER) proteins for degradation. (**A**) The degradation of an endogenous secretory protein with a C-terminal Flag (ER-Suc2-Flag) containing either DegV1 or ConV1 was monitored following addition of cycloheximide (CHX), using SDS-PAGE and immunoblotting. Loading controls were visualized by stain-free technology. (**B**) The degradation of a single transmembrane segment ER resident protein (Big1) with the N-terminus in the ER lumen, appended with either DegV1 or ConV1, was followed as in (**A**). (**C**) The degradation of polytopic integral membrane ER resident protein (Elo1) with the N-terminus in the ER lumen, appended with either DegV1 or ConV1, was followed as in (**A**). (**D**) The degradation of Big1 with DegV1 or ConV1 was followed as in (**A**) but following a 2 hr pretreatment with either ethanol (EtOH) or bortezomib (Btz) in a *pdr5Δ* strain. (**E**) The degradation of Elo1 with DegV1 or ConV1 was followed as in (**D**). All panels in this figure are representative of at least three independent biological replicates.

The online version of this article includes the following source data for figure 3:

**Source data 1.** Uncropped and labeled gels for *Figure 3*.

**Source data 2.** Raw unedited gels for *Figure 3*.

*and J*). Conversely, the Doa10 ubiquitin ligase was not required (*Figure 2I and J*). Consistently, degradation of each of DegV1-containing proteins were inhibited by either treating cells with bortezomib or genetic deletion of Hrd1.

When DegV1-containing proteins were targeted to the ER lumen, Hrd1-ERAD was required for their degradation (*Figure 2E–J*). We tested whether DegV1 would also function as a degron in the cytosol by removing the signal sequence. Somewhat surprisingly, we found that DegV1 also mediated proteasome-dependent degradation when localized to the cytosol (*Figure 2—figure supplement 2*). However, in contrast to ER-localized DegV1, when DegV1-containing proteins were localized to the cytosol, Hrd1 was not required for their degradation through the proteasome (*Figure 2—figure supplement 2*).

In these experiments, we confirmed that DegV1 targets heterologously expressed ER luminal proteins for ERAD-mediated proteasomal degradation. We next tested whether DegV1 could target endogenous *S. cerevisiae* proteins for degradation. We transplanted DegV1 onto three different classes of endogenous, ER-localized proteins. First, we used the endogenous protein Suc2. We used the Suc2 signal sequence followed by either DegV1, or ConV1, an HA tag, the Suc2 coding sequence, a Flag tag, and an HDEL (*Figure 3A*). With ConV1, Suc2 was stable over several hours, and, based on the modified glycosylation pattern, even appeared to be partially trafficked from the ER. In contrast, with DegV1 Suc2 was dramatically destabilized (*Figure 3A*). Next, we attached DegV1 to a type I integral membrane ER protein, called Big1, that has its N-terminus in the ER lumen and contains a single transmembrane segment (*Azuma et al., 2002*). We replaced the signal sequence of Big1 with the signal sequence of mating factor alpha followed by either DegV1 or ConV1, an HA tag, and the Big1 coding sequence. DegV1 was also able to destabilize the integral membrane protein Big1 (*Figure 3B*). Finally, we attached DegV1 or the control sequence to the N-terminus of Elo1, a multi-spanning integral membrane protein with seven probable transmembrane segments and the N-terminus in the ER

lumen (*Toke and Martin, 1996*; *Nie et al., 2021*). We found that DegV1 was capable of driving degradation for the multi-spanning membrane protein Elo1 (*Figure 3C*). These results support that DegV1 functions as an N-terminal degron for endogenous proteins with a range of topologies.

Soluble, luminal DegV1-containing proteins are targeted to the proteasome by the Hrd1-ERAD pathway (*Figure 2E–J*). To test whether DegV1-containing integral membrane proteins are also degraded by the proteasome, we followed Big1 and Elo1 degradation after treatment with cycloheximide and bortezomib. Both membrane proteins were significantly stabilized upon treatment with bortezomib (*Figure 3D and E*). Therefore, DegV1 targets both luminal and integral membrane ER proteins for recognition by ERAD and subsequent degradation by the proteasome.

## DegV1 is a functional degron in mammalian cells

DegV1 is a degron facilitating degradation from the ER lumen and also represents the first short, portable degron tag (<180 amino acids; *Carvalho et al., 2010*) identified for the Hrd1-ERAD system. We turned our attention to the possibility of using DegV1 as a tool in mammalian cells. To determine whether DegV1 functioned as an ER degron in mammalian cells, we generated an ER-targeted mNeonGreen (*Shaner et al., 2013*) by appending an N-terminal BiP signal sequence, the HA epitope tag,

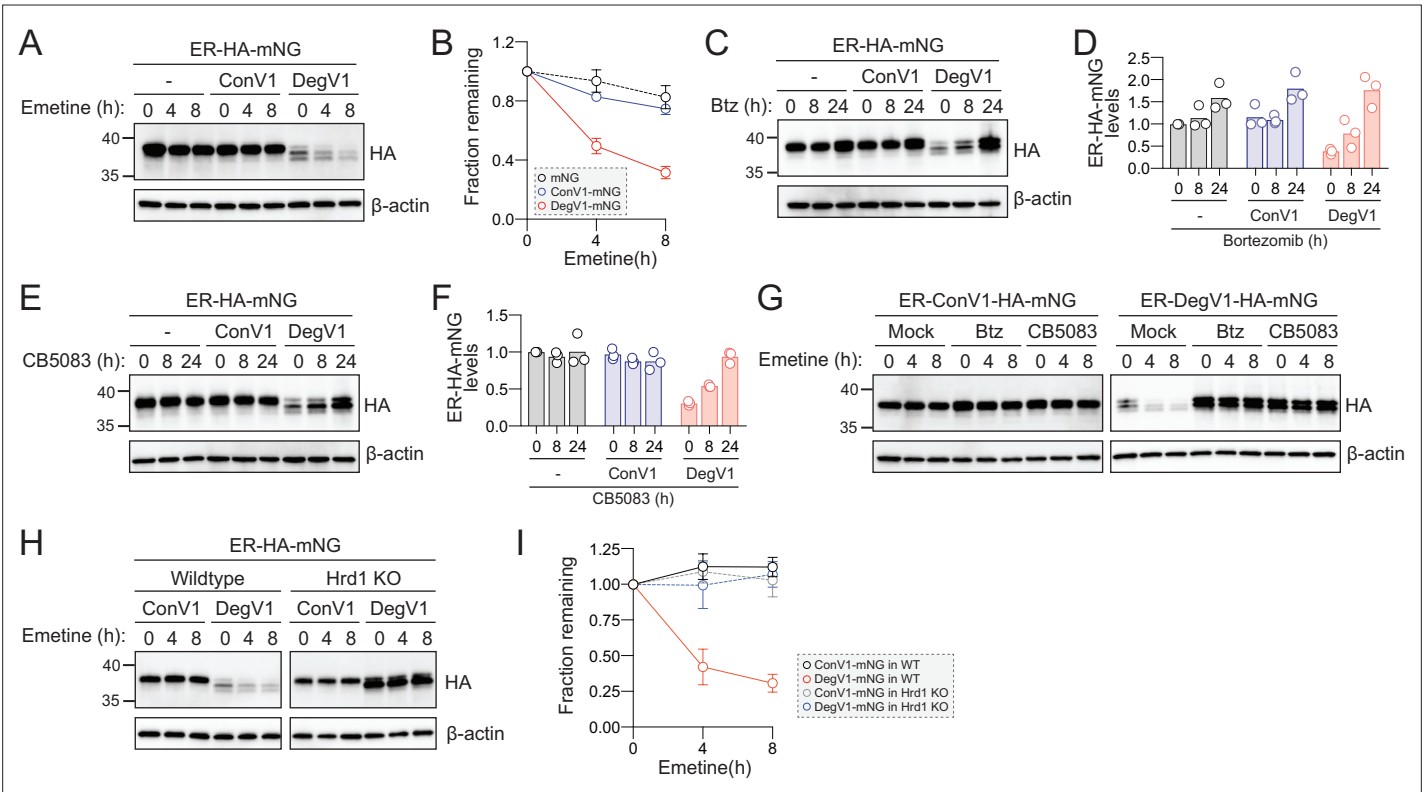

**Figure 4.** DegV1 functions as a degron in mammalian cells. (**A**) Endoplasmic reticulum (ER)-targeted mNeonGreen (ER-HA-mNG) was expressed alone (-), with ConV1, or DegV1 in U-2 OS cells by transient transfection. The degradation of ER-mNG was followed by immunoblotting with anti-HA antibody after treatment with 50 μM emetine. β-Actin was used as a loading control. (**B**) Anti-HA band intensities from (**A**) were quantified and normalized to the corresponding β-actin level. (**C**) As in (**A**) but after treatment with 50 nM bortezomib (Btz) for the indicated times. (**D**) Quantification of (**C**) normalized to the control protein (ER-HA-mNG). (**E**) As in (**A**) but after treatment with 1 μM CB5083, a p97 inhibitor, for the indicated times. (**F**) Quantification of (**E**) normalized to the control protein (ER-HA-mNG). (**G**) ER-HA-mNG with either ConV1 (left panel) or DegV1 (right panel) were expressed in U-2 OS pretreated with either 50 nM Btz or 1 μM CB5083 for 16 hr prior to an emetine chase. (**H**) The degradation of ER-HA-mNG with either ConV1 or DegV1 was followed in HEK293T cells or *HRD1⁻ᐟ⁻* cells using an emetine chase. (**I**) Quantification of (**H**). All panels in this figure are representative of at least three independent biological replicates and the quantification is presented as the mean ± standard deviation.

The online version of this article includes the following source data for figure 4:

**Source data 1.** Uncropped and labeled gels for *Figure 4*.

**Source data 2.** Raw unedited gels for *Figure 4*.

**Source data 3.** Numerical source data for plots displayed in *Figure 4*.

and the C-terminal ER retention peptide (KDEL) (ER-HA-mNG). We transfected U2OS cells with the ER-mNG containing either ConV1 or DegV1. The addition of DegV1, but not ConV1, reduced the steady-state ER-mNG levels compared to the control (*Figure 4A*, compare lanes 1, 4, and 7). When we inhibited translation with emetine, we found that ER-mNG and ER-ConV1-mNG were quite stable (*Figure 4A and B*). In contrast, ER-DegV1-mNG was unstable, with a half-life of approximately 4 hr (*Figure 4A and B*).

In *S. cerevisiae*, DegV1 degradation was dependent on the proteasome and mediated through ERAD. In U-2 OS cells, we tested whether degradation of DegV1 was also proteasome-dependent. We treated cells with bortezomib and found that, after an 8 hr treatment, ER-mNG and ER-ConV1-mNG levels remained largely unchanged, but after 24 hr we observed an ~50% increase. In contrast, for ER-DegV1-mNG proteasomal inhibition resulted in a dramatic accumulation of protein within 8 hr of treatment (*Figure 4C and D*). Remarkably, at 24 hr of bortezomib treatment ER-DegV1-mNG accumulated to similar levels compared to ER-ConV1-mNG, suggesting that the low steady-state level of ER-DegV1-mNG was caused by continuous degradation, rather than a general expression problem.

ERAD-dependent proteasomal degradation also requires the AAA-ATPase p97/VCP (Cdc48 in yeast). Therefore, we tested the stability of the ER-mNG proteins after treatment with the VCP inhibitor CB5083 (*Anderson et al., 2015*). Similar to the bortezomib treatment, we found that treatment with CB5083 resulted in stabilization of ER-DegV1-mNG (*Figure 4E and F*). As the CB5083 incubation length increased, we found that the levels of ER-DegV1-mNG approached those of the stable, control proteins. To confirm that bortezomib and CB5083 were preventing the active degradation of ER-DegV1-mNG, rather than just improving expression, we analyzed the degradation of the ER-mNG proteins in the presence of bortezomib or CB5083. We confirmed that, without addition of bortezomib or CB5083, ER-ConV1-mNG was stable while ER-DegV1-mNG was degraded (*Figure 4G*). When we pretreated cells with either bortezomib or CB5083 and inhibited translation with emetine, we found that the degradation of ER-DegV1-mNG was completely inhibited. Finally, we tested whether Hrd1 was required for degradation of ER-DegV1-mNG in mammalian cells. Using either wild-type or Hrd1 knockout cells (*Shi et al., 2017*), we followed degradation of ER-DegV1-mNG and found that the degron-containing protein was stabilized in the absence of Hrd1 (*Figure 4H and I*). Taken together, these data indicate that DegV1 functions as an Hrd1-dependent ER-localized degron in mammalian cells.

## Discussion

In this study, we identified the first short, linear, degron motifs that target proteins for degradation from the ER (*Figure 1*). We focused on functionalizing these motifs and found that DegV1 works with folded, luminal, and completely soluble proteins as well as integral membrane proteins with differing topologies. It works with both exogenous and endogenous proteins that are otherwise stable. This degron appears to only work at the N-terminus of nascent proteins, rather than internally or C-terminally. Importantly, DegV1 is degraded through the Hrd1-ERAD axis using the proteasome (*Figures 2 and 3*). Furthermore, we found that DegV1 works across eukaryotes in the mammalian cell culture system (*Figure 4*). Our data highlight a robust, highly potent degron that can be used for targeted protein degradation from the ER lumen and membrane.

Despite a growing understanding of how the ERAD system functions, the fundamental question of how degrons are recognized by ERAD remains unanswered. Here, we identified a number of different degrons, that remain to be completely characterized, but focused on a single degron (DegV1). It is somewhat surprising that DegV1 seems to only work as an N-terminal degron, but it is possible that when DegV1 is positioned anywhere except the N-terminus, DegV1 could be buried within other parts of the proteins. Although this position-dependent effect is simply a result of the methods used to identify this particular degron, it highlights the importance of future work to identify degrons capable of acting in different positions within a protein (amino terminal, internal, and carboxyl terminal). However, the sequence 'IHPYW' forming the basis of DegV1 appears to be relatively uncommon in nature and is not present in other proteins encoded in the *S. cerevisiae* genome. In fact, DegV1 appears to be absent from the currently annotated and available fungal genomes in the Saccharomyces Genome Database (*Cherry et al., 2012*). Based on our selection criteria, IHPYW alone is unlikely to be the most potent ER degron. In fact, we were able to demonstrate that just by increasing the length of our short degron, we could improve the degron to be more like that of a full ERAD substrate protein. We expect

that in most unfolded proteins, multiple short linear degrons would contribute to effective recognition and degradation by quality control systems.

Even with the limitations, we were able to demonstrate the utility of DegV1 and future degrons by illuminating that degrons can function even in evolutionarily distant species (*S. cerevisiae* and *Homo sapiens*). This was somewhat surprising because specific degrons transferred between fungi and animals are not always functionally conserved (*Timms et al., 2019*). It should be noted that the degradation rates of our degron-containing proteins are similar to the rates of other well-characterized ERAD substrates. We interpret these results to mean that DegV1 is degraded in a manner consistent with that of endogenous ERAD substrates. The retrotranslocation process itself appears to function by unfolding, or mostly unfolding, its substrates. Few protein transport systems can transport fully folded proteins across a lipid bilayer with the notable exceptions being the twin arginine transporter system (for review, see *Berks, 2015*) and peroxisomal import machinery (*Glover et al., 1994*; *McNew and Goodman, 1994*; *Romano et al., 2019*; *Gao et al., 2022*; *Skowyra and Rapoport, 2022*). However, previous studies in the ERAD field provide conflicting accounts over the ability to transport fully folded proteins from the ER to cytosol (*Tirosh et al., 2003*; *Bhamidipati et al., 2005*; *Shi et al., 2019*). What is certain is that glycosylated ERAD substrate proteins are retrotranslocated across the ER membrane, with the N-linked glycans representing a similar steric challenge compared to secondary structure or smaller folded proteins (*Grotzke et al., 2013*). Because DegV1 allows degradation of fully folded proteins, it is possible that the proteins are transported in a fully folded state. Although, it seems unlikely that a single heterodimeric channel (*Wu et al., 2020*) or homodimeric channel (*Schoebel et al., 2017*) would be able to transport a fully folded substrate. Perhaps, a substrate could occupy multiple ERAD complexes, could require other unidentified proteins, or could use mechanics similar to the peroxisomal machinery (*Gao et al., 2022*). However, we favor the idea that the targeted proteins are likely recognized and transported in an unfolded state after ERAD complex engagement. Future studies are needed to identify and characterize the molecular mechanisms that underpin this unexpected observation and it will be important to understand whether the core ERAD machinery, ER chaperones, or perhaps additional unidentified components are required.

Here, we have created the first tool that can be used to begin to answer these outstanding questions. We've identified a genetically encoded sequence that is easily manipulatable for targeting proteins for degradation from a previously unreachable cellular localization. The approach we've used will, eventually, enable targeted protein degradation from the ER to target physiologically, or pathophysiologically, relevant proteins. We anticipate that this platform will allow targeting and manipulation of a wide range of previously inaccessible proteins, not only for basic research, but for translational studies and, eventually, biomedical therapies.

# Methods

## Key resources table

| Reagent type (species) or resource | Designation | Source or reference | Identifiers | Additional information |
|---|---|---|---|---|
| Gene (*S. cerevisiae*) | HRD1 | Saccharomyces Genome Database (SGD) (*Wong et al., 2023*) | YOL013C | |
| Gene (*S. cerevisiae*) | HRD3 | Saccharomyces Genome Database (SGD) (*Wong et al., 2023*) | YLR207W | |
| Gene (*S. cerevisiae*) | USA1 | Saccharomyces Genome Database (SGD) (*Wong et al., 2023*) | YML029W | |
| Gene (*S. cerevisiae*) | DER1 | Saccharomyces Genome Database (SGD) (*Wong et al., 2023*) | YBR201W | |
| Gene (*S. cerevisiae*) | YOS9 | Saccharomyces Genome Database (SGD) (*Wong et al., 2023*) | YDR057W | |

*Continued on next page*

*Continued*

| Reagent type (species) or resource | Designation | Source or reference | Identifiers | Additional information |
|---|---|---|---|---|
| Gene (*S. cerevisiae*) | UBC7 | Saccharomyces Genome Database (SGD) (*Wong et al., 2023*) | YMR022W | |
| Gene (*S. cerevisiae*) | DOA10 | Saccharomyces Genome Database (SGD) (*Wong et al., 2023*) | YIL030C | |
| Gene (*S. cerevisiae*) | PDR5 | Saccharomyces Genome Database (SGD) (*Wong et al., 2023*) | YOR153W | |
| Gene (*S. cerevisiae*) | SUC2 | Saccharomyces Genome Database (SGD) (*Wong et al., 2023*) | YIL162W | |
| Gene (*S. cerevisiae*) | BIG1 | Saccharomyces Genome Database (SGD) (*Wong et al., 2023*) | YHR101C | |
| Gene (*S. cerevisiae*) | ELO1 | Saccharomyces Genome Database (SGD) (*Wong et al., 2023*) | YJL196C | |
| Strain, strain background (*S. cerevisiae*) | For strains, see *Supplementary file 1* | This study | NA | For strains, see *Supplementary file 1* |
| Cell line (*H. sapiens*) | HEK293 | *Shi et al., 2017* | NA | Ling Qi lab (University of Virginia) |
| Cell line (*H. sapiens*) | HEK293 Hrd1 Knockout | *Shi et al., 2017* | NA | Ling Qi lab (University of Virginia) |
| Cell line (*H. sapiens*) | U-2 OS | ATCC | ATCC HTB-96 | |
| Transfected constructs (*H. sapiens*) | For plasmids, see *Supplementary file 2* | This study | NA | For plasmids, see *Supplementary file 2* |
| Transfected constructs (*S. cerevisiae*) | For plasmids, see *Supplementary file 2* | This study | NA | For plasmids, see *Supplementary file 2* |
| Recombinant DNA reagent | For plasmids, see *Supplementary file 2* | This study | NA | For plasmids, see *Supplementary file 2* |
| Sequence-based reagent | For primers, see *Supplementary file 3* | This study | NA | For primers, see *Supplementary file 3* |
| Antibody | THE DYKDDDDK Tag Antibody, mAb (mouse monoclonal) | GenScript | A00187; RRID:AB_1720813 | Used at 1:2000 dilution |
| Antibody | Anti-GFP (rabbit polyclonal) | GenScript | A01704; RRID:AB_2622199 | Used at 1:2000 dilution |
| Antibody | Anti-HA High Affinity antibody (clone 3F10) (rat monoclonal) | Roche | 11867423001; RRID:AB_390918 | Used at 1:2500 dilution |
| Antibody | THE V5 Tag Antibody, mAb, (mouse monoclonal) | GenScript | A01724; RRID:AB_2622216 | Used at 1:2500 dilution |
| Antibody | Amersham ECL Rat IgG, HRP-linked whole antibody (from goat) (polyclonal secondary) | Cytiva | NA935; RRID:AB_772207 | Used at 1:4000 dilution |
| Antibody | Amersham ECL Rabbit IgG, HRP-linked whole Ab (from donkey) (polyclonal secondary) | Cytiva | NA934; RRID:AB_772206 | Used at 1:4000 dilution |
| Antibody | Amersham ECL Mouse IgG, HRP-linked whole Ab (from sheep) (polyclonal secondary) | Cytiva | NA931; RRID:AB_772210 | Used at 1:4000 dilution |

*Continued on next page*

*Continued*

| Reagent type (species) or resource | Designation | Source or reference | Identifiers | Additional information |
|---|---|---|---|---|
| Antibody | Goat anti-Mouse IgG (H+L) Highly Cross-Adsorbed Secondary Antibody, Alexa Fluor Plus 800 (polyclonal secondary) | Invitrogen | A32730; RRID:AB_2633279 | Used at 1:4000 dilution |
| Antibody | Anti-β-actin | Cell Signaling | NA | |
| Peptide, recombinant protein | Phusion High-Fidelity DNA Polymerase | New England Biolabs | M0530S | |
| Peptide, recombinant protein | Zymolyase 100T | AMSBIO | 120493-1 | |
| Commercial assay, kit | ECL Select Western Blotting Detection Reagent | Cytiva | RPN2235 | |
| Commercial assay, kit | BCA assay | Thermo Fisher Scientific | 23225 | |
| Commercial assay, kit | NEBuilder HiFi DNA Assembly Master Mix | New England Biolabs | E2621 | |
| Commercial assay, kit | QIAquick PCR Purification Kit | QIAGEN | 28104 | |
| Commercial assay, kit | Invitrogen dsDNA HS assay | Invitrogen | Q32854 | |
| Chemical compound, drug | Cycloheximide | Sigma-Aldrich | 239763 | |
| Chemical Compound, drug | Emetine | Calbiochem | 324693 | |
| Chemical compound, drug | CB-5083 | Cayman Chemicals | 19311 | |
| Chemical compound, drug | Bortezomib | APExBIO | A2614 | |
| Chemical compound, drug | SYTOX Blue Nucleic Acid Stain - 5 mM Solution in DMSO | Invitrogen | S11348 | |
| Chemical compound, drug | Invitrogen UltraPure Salmon Sperm DNA Solution | Thermo Fisher Scientific | 15061 | |
| Chemical compound, drug | Fetal Bovine Serum, Regular, USDA Approved Origin | Corning | 35-010-CV | |
| Chemical compound, drug | DMEM (Dulbecco's Modified Eagle's Medium) | Corning | 10-013-CV | |
| Chemical compound, drug | Lipofectamine 2000 Transfection Reagent | Invitrogen | 11668019 | |
| Chemical compound, drug | DMSO | Sigma-Aldrich | D2650 | |
| Software, algorithm | ImageJ | *Schneider et al., 2012* | RRID:SCR_003070 | |
| Software, algorithm | ImageLab version 6.1 | Bio-Rad | https://www.bio-rad.com/en-us/product/image-lab-software?ID=KRE6P5E8Z | |
| Software, algorithm | FlowJo version 10.7.1 | Becton, Dickinson and Company | https://www.flowjo.com/solutions/flowjo | |
| Software, algorithm | GraphPad Prism | Dotmatics | https://www.graphpad.com/ | |
| Software, algorithm | Illustrator | Adobe | https://www.adobe.com/products/illustrator.html | |

## Yeast strains and plasmids

Yeast were cultured at 30°C in synthetic complete medium (SC) supplemented with the appropriate amino acids. The *hrd1Δ* and *pdr5Δ* strain used in this study were derivatives of BY4741 (MATa *his3Δ1*

*leu2Δ0 met15Δ0 ura3Δ0*) or BY4742 (*MATα his3Δ1 leu2Δ0505lys2Δ0 ura3Δ0*). The *hrd1Δpdr5Δ* strain was generated by crossing *hrd1Δ* and *pdr5Δ* strains, sporulating the diploids, and screening the appropriate loci by PCR. For a list of yeast strains used in this study, see *Supplementary file 1*. For a list of plasmids used in this study, see *Supplementary file 2*. Plasmids were constructed using restriction enzyme cloning or NEB HiFi assembly. Plasmids used in this study were either centromeric (*Sikorski and Hieter, 1989*) or custom integrating plasmids (*Hwang et al., 2023*). For a list of primers used to generate the pentapeptide library, see *Supplementary file 3*. Plasmids were transformed into yeast using the LiAc/PEG method (*Gietz and Schiestl, 2007*). Following transformation into yeast, three to four independent transformants were passaged one to two times on selection media before using in experiments.

## Mammalian cell culture and transfection

U-2 OS and HEK293 cells were cultured in DMEM containing 4.5 g/L glucose and L-glutamine, and supplemented with 10% fetal bovine serum (Corning) at 37°C and 5% $CO_2$. Cells at 60–80% confluence were transiently transfected with the indicated plasmids using Lipofectamine 2000 (Invitrogen, 11668019) according to the manufacturer's protocols. After 24 hr, cells were split for emetine chase or chemical treatment assays.

## Mammalian cell lysis

Cells were treated with a translation inhibitor, 50 μM emetine (Calbiochem, 324693) for the indicated time periods. For chemical treatments, cells were either mock treated with DMSO, treated with 50 nM bortezomib (APExBIO, A2614), or treated with 1 μM CB5083 (Cayman Chemicals, 19311) for the indicated time periods prior to collection. For combination treatments, cells were pretreated with either bortezomib, or CB5083, for 16 hr prior to emetine chase. The cells were collected and washed once in 1× PBS (0.01 M phosphate buffered saline, pH 7.4, 0.138 M NaCl, 0.0027 M KCl) before lysis in 50 mM Tris, pH 7.4, 150 mM NaCl, 1% Triton X-100, 1 mM PMSF, and protease inhibitor cocktail for 10–20 min at 4°C. The lysates were cleared by centrifugation at 20,000×*g* for 10 min at 4°C. Protein concentrations were determined using a BCA assay (Thermo Fisher Scientific, 23225). Cell lysates were normalized to the same concentrations in Laemmli sample buffer and heated to 65°C for 5 min prior to separation by SDS-PAGE, transferred to a PVDF membrane, immunoblotted with antibodies (anti-HA from Roche, anti-β-actin from Cell Signaling, HRP-linked ECL rabbit-IgG and rat-IgG from Cytiva), and detected by chemiluminescence (ECL Select Western blotting detection reagent, Cytiva) using a ChemiDoc MP (Bio-Rad). For quantification of the immunoblot band intensities, we used ImageLab version 6.1 (Bio-Rad). Band intensities were normalized to β-actin protein in the sample quantified within each lane.

## Yeast pentapeptide library generation

We designed an in vivo gap repair strategy for cloning our pentapeptide libraries into the ER-tFT. The plasmid backbone was based on pRS416 (*Christianson et al., 1992*) and contained a *TDH3* promoter (also known as GPD or GAPDH), the signal sequence from mating factor alpha, mCherry, GFP, an HDEL ER retention signal, and the *CYC1* terminator. To prevent the peptide library from potentially disrupting signal sequence cleavage, two amino acids (Ala and Ser) after the signal sequence cleavage site were left upstream of the library. The final N-terminal amino acid sequence of the ER-tFT library after translocation and signal peptide cleavage is ASXXXXX.

To generate the pentapeptide DNA library fragment with homology arms to ER-tFT, four PCRs were performed (*Figure 1—figure supplement 1C*) with Phusion polymerase (New England Biolabs, M0530S). We started by generating a linear DNA template to reduce bias in our PCRs. Using pRP01 and primers prRP07 and prRP08, we amplified a 1025 bp fragment with 524 bp upstream (overlapping the *TDH3* promoter and signal sequence) and 495 bp downstream (overlapping mCherry) of the library insertion (EcoRI) site in pRP01. This fragment (fragment 1) was gel-purified (QIAGEN, 28104) to remove any residual plasmid and contain only a linear template.

Fragment 2, containing the upstream homology arm, was generated through PCR using fragment 1 (the linear DNA template) with primers prRP10 and prRP29 to amplify a 512 bp fragment, which included homology with both the *TDH3* promoter and signal sequence and contained the random DNA library insertions. Fragment 3, containing the downstream homology arm, was generated in a

PCR using fragment 1 (the linear DNA template) from the first PCR with primers prRP09 and prRP51 to generate a 146 bp fragment, which included random DNA in the library position along with homology arms in mCherry. Only 102 bp of homology was included in the mCherry coding region to limit the number of mutations found in mCherry included by homologous recombination. When we included longer homology arms into mCherry, we found that our screening procedure was selective enough to identify mutations in mCherry which could lead to false 'unstable' hits (data not shown).

To generate the PCR product for homologous recombination containing overlapping homology arms to the target plasmid (pRP01), we gel-purified fragments 2 and 3 and mixed the DNA in an equimolar ratio. Using primers prRP29 and prRP51, we used 25 PCR cycles to generate a 605 bp fragment 4 (containing the pentapeptide library and homology arms covering 488 bp upstream and 111 bp downstream of the library cut site). Fragment 4 was purified and mixed at a 30:1 molar ratio with purified pRP01 (digested with EcoRI) immediately before yeast electroporation.

The library transformation into yeast strain yRB203 was performed as previously described (*Benatuil et al., 2010*). Cells were grown overnight to stationary phase in YPD media, shaking at 225 rpm and 30°C. An aliquot of the overnight culture was used to inoculate 400 mL of YPD media at 0.3 $OD_{600}$/mL. Cells were grown for approximately 5 hr until 1.6 $OD_{600}$/mL was reached and collected by centrifugation at 3200×$g$ for 5 min. The cell pellet was washed twice by 200 mL of ice-cold water and once by 200 mL of electroporation buffer (1 M sorbitol/1 mM $CaCl_2$, sterile filtered). The cell pellet was then resuspended in 80 mL of 100 mM LiAc/10 mM DTT, split into two aliquots of 40 mL, and each was incubated in a 250 mL culture flask for 30 min at 30°C, shaking at 225 rpm. Next, cells were collected by centrifugation, washed once with 200 mL of ice-cold electroporation buffer, and resuspended to 2.4 mL in electroporation buffer. The cell resuspension was evenly divided into six pre-chilled Bio-Rad Gene Pulser cuvettes (0.2 cm electrode gap) and kept on ice for 10 min with DNA. One reaction was used as a no DNA control, one reaction received digested vector only, and four cuvettes received 3 µg of digested vector and 9 µg fragment 4. Cells were electroporated at 2.5 kV and 25 µF, with time constants varying from 4.0 to 4.3 ms. Cells were gently transferred from each cuvette into 8 mL of a 1:1 mix of 1 M sorbitol:YPD in culture tubes (25 m diameter) and incubated at 30°C, with shaking at 220 rpm. After 1 hr, cells were pelleted by centrifugation and inoculated into 1 L SC dropout media (-ura) at 0.2 $OD_{600}$/mL. Dilutions from the electroporated cells were also plated on SC dropout plates and grown for 2 days at 30°C to determine the library transformation size (approximately 1.2 million for the library described here).

Electroporated cells were grown for ~18 hr while shaking at 30°C until reaching 0.5 $OD_{600}$/mL. Control strains expressing ER-tFT (pRP01) and KHN-tFT (pRP08) were also cultured in parallel. The strains and library were then treated with DMSO only (Sigma-Aldrich, D2650) or 50 µg/mL cycloheximide (EMD Millipore, 239763) for 2 hr. After treatment, the cells were pelleted, washed once in 1× PBS, resuspended in 1× PBS containing 1 µM Sytox Blue (Invitrogen, S11348), and incubated at 4°C prior to cell sorting.

## Fluorescence-activated cell sorting

Cells were sorted on a MoFlo Astrios Cell Sorter (Beckman Coulter) running Summit software. The instrument was set with a 100 µm tip, 405 nm laser with 448/59 nm bandpass filter, 488 nm laser with 514/20 nm bandpass filter, and 561 nm laser with 620/29 nm bandpass filter. Events were gated to select for yeast cell-sized events, single cells, live cells, and mCherry-GFP positive cells (*Figure 1—figure supplement 1A*). The mCherry/GFP ratio for each cell in the final gated population was displayed as a histogram. The ER-tFT and KHN-tFT controls were used to help define where to draw the low mCherry/GFP ratio (unstable) bin for sorting (*Figure 1—figure supplement 1D*). In total, 13 million mCherry-GFP positive events were sorted from the pentapeptide-ER-tFT library, >10 times over the library size. The sorted cells were grown at 30°C with shaking in 5 mL of SC dropout media for 24 hr and expanded to 25 mL cultures overnight. 10 $OD_{600}$ of both the 'unstable' bin and unsorted pentapeptide-ER-tFT library were pelleted and flash-frozen in liquid nitrogen and stored at –80°C prior to DNA extraction.

## DNA extraction and amplicon sequencing preparation

DNA extraction was performed essentially as previously described (*Kats et al., 2018*). The frozen 10 $OD_{600}$ pellets were resuspended in 500 µL of 10 mM $K_2HPO_4$ pH 7.2, 10 mM EDTA, 50 mM

2-mercaptoethanol and incubated with 50 mg/mL zymolyase 100T (AMSBIO) at 37°C for 30–60 min until the mixture became clear. 100 µL of lysis buffer (25 mM Tris-HCl pH 7.5, 25 mM EDTA, 2.5% SDS [wt/vol]) was added and the suspension was incubated at 65°C for 45 min. Proteins were precipitated by adding 166 µL of 3 M potassium acetate and incubating on ice for 10 min. Samples were then centrifuged at 21,000×$g$ for 10 min at 4°C. The supernatant containing DNA was collected and the DNA was precipitated by the addition of 800 µL of 100% ethanol, followed by centrifugation at 21,000×$g$ for 10 min at 4°C. Precipitated DNA was washed with 70% (vol/vol) ethanol and resuspended in 80 µL of water.

Next, 5 µL of the isolated DNA solution to amplify a 217 bp fragment encompassing the pentapeptide sequences. Partial adapters for Illumina sequencing were added by 25 cycles of PCR using primers prRP37 and prRP38 (annealing temperature of 60°C using Phusion DNA polymerase). PCR products were purified (QIAGEN, 28104), normalized to 20 ng/µL using a QuBit 3 (Invitrogen, Q32854), and sent for amplicon sequencing using Genewiz Amplicon-EZ (now Azenta Life Sciences).

## Amplicon-EZ analysis

Sequences from the two Amplicon-EZ samples (unstable bin and input library) were analyzed for quality, trimmed, aligned, and translated by Genewiz (now Azenta Life Sciences). The sequences and translations can be found in *Source data 1*, sheets labeled Library Data and Sorted Bin Data. From the translated sequences, the pentapeptide region was isolated and divided into positions #1–5 (P1-P5). The amino acid count was the sum of occurrences for each amino acid at each position (*Source data 1*, AA Analysis Sheet). We then divided the amino acid frequency at each position for the library or unstable sorted bin by the expected amino acid frequency based on the number of codons that encode a given amino acid. This gave us the relative enrichment of each amino acid at each of the five positions (*Figure 1F*).

## Flow cytometry-based degradation assays

For each experiment, two biological replicates were transferred into SC dropout media in a 96-well plate (Fisherbrand, 12566611) sealed with gas-permeable membranes (Sigma-Aldrich, Z763624) and grown overnight shaking at 1000 rpm at 30°C. Overnight cell density was typically around ~4–5 OD$_{600}$/mL. In the morning, cells were diluted to 0.2 OD$_{600}$/mL in SC dropout media and grown at 30°C shaking at 1000 rpm for ~5 hr or until the OD$_{600}$/mL of the cultures was ≥0.5. Cells were pelleted at 3200×$g$ and the supernatant was removed by aspiration prior to resuspension in SC dropout media at 2 OD$_{600}$/mL. For experiments using cells treated with 50 µg/mL cycloheximide or 50 µM bortezomib, cells were transferred directly into new 96-well plates (Grenier Bio-One, 650185). To account for slowed growth by these treatments, in the same plates the untreated/DMSO-treated cells were diluted by ⅓ with fresh media. During the treatment periods, cells were incubated at 30°C while shaking at 600 rpm. After treatment, cells were pelleted at 3200×$g$, washed once with 1× PBS, and resuspended in 1× PBS with 1 µM Sytox Blue (Invitrogen). Cells were maintained at 4°C during flow cytometry analysis on a MACSQuant VYB (Miltenyi) running MACSQuantify software (version 2.13.2). Sytox Blue was followed using the 405 nm laser and 452/45 nm emission filters. SuperfastGFP was followed using the 488 nm laser and 452/45 nm emission filters. mCherry was followed using the 561 nm laser 615/20 nm emission filters. Downstream analyses were performed in FlowJo (version 10.7.1) with event gating to select yeast cell-sized events, single cells, live cells, and mCherry-GFP positive cells (*Figure 1—figure supplement 1A*). The mCherry/GFP ratio of mCherry-GFP positive cells was calculated in FlowJo and analyzed using a one-way ANOVA and Tukey's multiple comparisons tests in GraphPad Prism.

## Immunoblotting-based degradation assays

Cycloheximide-chase degradation assays were performed as described previously (*Hwang et al., 2023*; *Peterson et al., 2023*) with the following modifications. Starter cultures were grown overnight in SC dropout media while shaking at 30°C. Cultures were diluted to 0.2 OD$_{600}$/mL in SC dropout media and grown for 4–5 hr to mid-log phase (0.4–1.0 OD$_{600}$/mL). Cultures were pelleted at 3200×$g$ for 5 min and resuspended to 2.0 OD$_{600}$/mL in fresh media before treatment with 50 µg/mL cycloheximide. Samples were incubated at 30°C with shaking and, at the indicated time points, shifted to 4°C and collected by centrifugation at 21,000×$g$ for 5 min. The supernatant was removed and cell pellets

were incubated on dry ice prior to storage at –80°C or cell lysis. For experiments with bortezomib treatment, cells were pretreated with 50 μM bortezomib for 15 min prior to cycloheximide addition.

Cells were resuspended in SUME lysis buffer (1% SDS, 8 M urea, 10 mM MOPS, pH 6.8, 10 mM EDTA) (*Gardner et al., 1998*) at 20 OD$_{600}$/mL with acid-washed glass beads (0.1 mm, Bio-Spec). Cells were vortexed for 2 min and an equal volume of sample buffer (4% SDS, 8 M urea, 125 mM Tris pH 6.8, 10% β-mercaptoethanol, 0.02% bromophenol blue) was added and briefly vortexed. The samples were incubated at 65°C for 5 min, separated by SDS-PAGE, transferred to a PVDF membrane, immunoblotted with antibodies (anti-GFP from GenScript, anti-DYKDDDK from GenScript, anti-HA from Roche, anti-V5 from GenScript, HRP-linked ECL rabbit-IgG and mouse-IgG from Cytiva, Goat anti-Mouse IgG Alexa Fluor Plus 800 from Invitrogen), and detected by chemiluminescence (ECL Select Western Blotting Detection Reagent, Cytiva) or by Dylight800 fluorescence using a ChemiDoc MP (Bio-Rad). For quantification of the immunoblot band intensities, we used ImageLab version 6.1 (Bio-Rad). Band intensities were normalized to total protein in the sample quantified within each lane using Stain-Free Dye Imaging (Bio-Rad).

## Materials availability statement

Further information and requests for resources should be directed to and will be fulfilled by the lead contact, Ryan Baldridge (ryanbald@umich.edu).

## Acknowledgements

We would like to thank Ling Qi for sharing the Hrd1-deficient HEK293 cells. We would also like to thank Andrew Folkmann and Kaushik Ragunathan for their critical reading of the manuscript, Basila Moochickal Assainar for help with the fluorescence microscopy, and other members of the Baldridge lab for their thoughtful discussion and comments regarding this work. RS (previously R Plumb) was supported by an NIH/NIGMS Award (5F32GM136020) and DD was supported by the NIH Cellular and Molecular Biology Training Grant (T32-GM007315). In addition, this work was supported by an NIH/NIGMS Award (R35GM128592 to RDB).

## Additional information

### Competing interests

Rachel Sharninghausen, Ryan D Baldridge: has filed a patent application (PCT/US2024/011152) related to the use of this technology for targeted protein degradation. The other authors declare that no competing interests exist.

### Funding

| Funder | Grant reference number | Author |
| --- | --- | --- |
| National Institute of General Medical Sciences | 5F32GM136020 | Rachel Sharninghausen |
| National Institute of General Medical Sciences | T32GM007315 | Devon D Dennison |
| National Institute of General Medical Sciences | R35GM128592 | Ryan D Baldridge |

The funders had no role in study design, data collection and interpretation, or the decision to submit the work for publication.

### Author contributions

Rachel Sharninghausen, Conceptualization, Formal analysis, Validation, Investigation, Visualization, Methodology, Writing – original draft, Writing – review and editing; Jiwon Hwang, Formal analysis, Validation, Investigation, Visualization, Methodology, Writing – review and editing; Devon D Dennison, Formal analysis, Validation, Investigation, Visualization, Writing – review and editing; Ryan D Baldridge, Conceptualization, Formal analysis, Supervision, Funding acquisition,

Investigation, Visualization, Writing – original draft, Project administration, Writing – review and editing

### Author ORCIDs
Ryan D Baldridge  https://orcid.org/0000-0001-7158-7812

Reviewer #1 (Public review): https://doi.org/10.7554/eLife.89606.3.sa1
Reviewer #2 (Public review): https://doi.org/10.7554/eLife.89606.3.sa2
Author response https://doi.org/10.7554/eLife.89606.3.sa3

## Additional files

### Supplementary files
• Supplementary file 1. Yeast strains used in this study.
• Supplementary file 2. Plasmids used in this study.
• Supplementary file 3. Primers used in this study.
• MDAR checklist
• Source data 1. Sequencing reads and translations. Illumina sequencing reads were trimmed, translated into amino acids, and analyzed as described in the Methods section.

### Data availability
All data generated or analysed during this study are included in the manuscript and supporting files.

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
