## [Editor Report · eLife Assessment]

This **important** study identifies a short amino acid sequence that, when fused in multimeric form to the amino termini of luminal ER proteins, initiates proteasomal degradation via the Hrd1 ER quality control ubiquitin ligase complex. The authors provide **solid** evidence that this sequence functions as a "degron" for ER proteins. Future work is required to obtain a more detailed view of the properties of this degron, the mechanisms underlying its recognition by ER-resident and cytoplasmic factors, and the in vivo relevance of the findings.

---

## [Referee Report · Reviewer #1 (Public review)]

The authors use a previously established reporter comprising a slow- and a fast-folding fluorescent protein fused to a randomly-generated library of penta-peptides at its amino-terminus and a signal sequence for import into the endoplasmic reticulum (ER). They then determine the stability of these constructs in a high throughput FACS-sorting procedure and identify a set of peptides that route the construct to proteasomal degradation. Increasing the copy number of one of these peptides further decreases the stability of the construct. This polypeptide resembles a "degron" for ER proteins, because it also targets other ER proteins with different topological and folding properties for degradation. It only works when placed at the amino-terminus of a protein and utilizes components of the Hrd1 ubiquitin ligase complex, a well-established quality control ubiquitin ligase in the ER membrane. Importantly, the degron also targets ER-proteins in mammalian cells.

The authors convincingly show that fusion of their newly identified degron to the amino terminus of ER-resident proteins with different topology suffices to target them for proteasomal degradation. The data for this are well-founded and contain appropriate controls. While technically sound, the study does only give superficial information on general properties of the degron and its recognition by cellular factors. Further simple experiments would have addressed a number of important points. The authors only provide data about the composition of the identified amino acid sections from the high-throughput approach and the statistical preference for certain amino acids at individual positions. They do not study degron composition experimentally by substituting individual amino acids with other residues and analyzing protein stability. Increasing the numbers of the initially identified degron pentamer increases substrate turnover, but the basis for this remains unclear. Each copy may be actively involved in better recognition, elongation of the degron may facilitate accessibility by recognition factors or multiplying the short amino acid stretch may generate new signatures at the amino-terminus that are more readily recognized by a quality control machinery. Consequently, this study does not allow conclusions to be drawn about general properties of degron composition and/or structure. The degron also functions with cytoplasmic proteins, suggesting that similar characteristics of a polypeptide attract the attention of quality control systems also in other cellular compartments. However, the authors did not pursue this finding further, e.g. by identifying factors for degron recognition in the cytoplasm. It would have been particularly interesting to test whether the degron would initiate degradation when placed at cytoplasmically-exposed amino termini of membrane-bound ER proteins. Information on degron properties is required to better understand principles of substrate recognition by protein quality control pathways and to design constructs for targeting endogenous proteins via proteolysis targeting chimeras (PROTACs).

---

## [Referee Report · Reviewer #2 (Public review)]

Summary:

Sharninghausen et al use a generic screening platform to search for short (5 amino acid) degrons that function in the lumen of the endoplasmic reticulum (ER) of budding yeast. The screen did indeed identify a number of sequences which increased the rate of degradation of their test proteins. Although the effect of the single degron was rather modest the authors could show that by mutimerising the sequence (4x) they obtained degrons that functioned fairly efficiently. Further characterisation indicated that the degrons only functioned when placed at the N-terminus of the target protein and, were dependent on both the proteasome and the segregase Cdc48 (p97) for degradation. The authors also demonstrated that degradation was via the ERAD pathway.

Strengths:

In general, the data presented is supportive of the conclusions drawn and the authors have thus identified a sequence that can be appended onto other ER targeted proteins to mediate their degradation within the lumen of the ER. How useful this will be to the community remains to be seen.

Weaknesses:

While the observation that such mutimerised sequences can act as degrons is an interesting curiosity, it is not clear that such sequences function in vivo. In fact the DegV1 sequence used throughout the paper is not present in any yeast or fungal proteins and the fact that it has to be located at the N-terminus of the protein to induce degradation is at odds with the idea that proteins to be degraded need to be unfolded. Thus, the role of such sequences in vivo is questionable.

Comments on revised manuscript:

Although the role of such degron sequences remains to be determined in vivo, it is clear that the authors have developed a tool that could be useful to the scientific community. The specific points raised were appropriately addressed by the authors.

---

## [Author Response]

The following is the authors’ response to the original reviews.

We would like to thank the reviewers for their constructive feedback and overall positive response to our manuscript. Reviewer #1 had no specific recommendations, so below we address Reviewer #2’s comments.

**Reviewer #2 (Recommendations For The Authors):**
Specific points(1) In Fig. 1H peptides selected are much more stable than the positive control KHN-FT, but they appear to be less stable than randomly selected 5 amino acid sequences. Are the differencesbetween the randomly selected sequences and the selected sequences statistically significant.

Thank you for the feedback. Yes, the differences are statistically significant by one-way ANOVA and the Tukey’s multiple comparisons tests, we’ve updated the figure legend to indicate this fact.

(2) In Fig. 1I the FACS profile of 4x looks like that of KHN, but it is very difficult to see in the figure. Looking at the quantitation in Fig. 1J it is impossible to compare KHN with 4x as the KHN is on the baseline. Could this be improved by using a log scale to present the data.

Thank you for pointing this out. We’ve improved the figure so the KHN is easier to see. In addition, we’ve attempted different way to display these results, but settled on scaling the data between 1 and 0 as our comparison points. We’ve updated the main figure to more clearly show this result so the KHN is easier to compare.

(3) In Fig. 2G and Fig. 2F don't really match up. It looks from Fig. 2G like there is still some degradation in the hrd1 deletion strain, but this is not reflected in the quantitation (Fig. 2H).

To our eyes, the degradation in a hrd1null appears to be quite small, which seems to be reflected in the quantification (~20% decrease over 90 minutes). We included the figure in Author response image 1 for quick comparison.

(4) Throughout the paper the authors claim that the proteins are degraded by a cytosolic proteasome. I agree that the proteins are degraded via the proteasome, but I don't see any evidence that it is cytosolic.

Thank you for pointing this out. We’ve adjusted the text to reflect the fact that the proteasomal degradation is not necessarily in the cytosol.